# Association of HbA1c with VO_2max_ in Individuals with Type 1 Diabetes: A Systematic Review and Meta-Analysis

**DOI:** 10.3390/metabo12111017

**Published:** 2022-10-24

**Authors:** Max L. Eckstein, Felix Aberer, Florian J. R. Dobler, Faisal Aziz, Tim Heise, Harald Sourij, Othmar Moser

**Affiliations:** 1BaySpo—Bayreuth Center of Sport Science, Research Group Exercise Physiology and Metabolism, University Bayreuth, 95447 Bayreuth, Germany; 2Division of Endocrinology and Diabetology, Medical University of Graz, 8036 Graz, Austria; 3Profil, 41460 Neuss, Germany

**Keywords:** type 1 diabetes, HbA_1c_, VO_2max_, systematic review, meta-analysis

## Abstract

The aim of this systematic review and meta-analysis was to evaluate the association between glycemic control (HbA_1c_) and functional capacity (VO_2max_) in individuals with type 1 diabetes (T1DM). A systematic literature search was conducted in EMBASE, PubMed, Cochrane Central Register of Controlled Trials, and ISI Web of Knowledge for publications from January 1950 until July 2020. Randomized and observational controlled trials with a minimum number of three participants were included if cardio-pulmonary exercise tests to determine VO_2max_ and HbA_1c_ measurement has been performed. Pooled mean values were estimated for VO_2max_ and HbA_1c_ and weighted Pearson correlation and meta-regression were performed to assess the association between these parameters. We included 187 studies with a total of 3278 individuals with T1DM. The pooled mean HbA_1c_ value was 8.1% (95%CI; 7.9–8.3%), and relative VO_2max_ was 38.5 mL/min/kg (37.3–39.6). The pooled mean VO_2max_ was significantly lower (36.9 vs. 40.7, *p* = 0.001) in studies reporting a mean HbA_1c_ > 7.5% compared to studies with a mean HbA_1c_ ≤ 7.5%. Weighted Pearson correlation coefficient was r = −0.19 (*p* < 0.001) between VO_2max_ and HbA_1c_. Meta-regression adjusted for age and sex showed a significant decrease of −0.94 mL/min/kg in VO_2max_ per HbA_1c_ increase of 1% (*p* = 0.024). In conclusion, we were able to determine a statistically significant correlation between HbA_1c_ and VO_2max_ in individuals with T1DM. However, as the correlation was only weak, the association of HbA_1c_ and VO_2max_ might not be of clinical relevance in individuals with T1DM.

## 1. Introduction

Regular physical activity (PA) represents a highly relevant non-pharmaceutical-glucose lowering activity in people with type 1 diabetes (T1DM) and is recommended to be conducted by means of moderate-to-vigorous intensity aerobic exercise for 150 min at least three days a week in most adults with T1DM [1]. Despite the challenge of PA potentially contributing to exercise-induced dysglycemia in T1DM, potential benefits have been demonstrated in regard to improving cardio-respiratory fitness, cardiovascular risk factors, and quality of life [2]. Besides the impact of PA on these aforementioned parameters, deteriorated glycemic control may act as an antagonist, which in T1DM remains a matter of debate [3] In people with type 2 diabetes (T2DM), research evidence suggests an inverse correlation between cardiorespiratory fitness assessed by maximum oxygen uptake (VO_2max_) and clinical markers of glycemic control (high HbA_1c_, high fasting glucose) also after adjustment for age, body weight and markers of adiposity [4]. It has been speculated that VO_2max_ as a determinant for PA may be positively associated with β-cell compensation mechanisms such as improvements in glucolipotoxicity [5]. This may lead to lower levels of pro-inflammatory cytokines and increased secretion of various growth factors and hormones which may contribute to higher β-cell mass [5]. However, these suggested mechanisms in T2DM are hardly transferable to a T1DM population that often has completely lost their endogenous insulin production during the progress of the autoimmune disease. 

We have recently shown that individuals with early-stage T1DM compared to healthy controls are characterized by a lower maximum oxygen uptake including decreased absolute and relative oxygen uptake (VO_2_) as well as weaker oxygen reserves and oxygen pulse when determined during a cardiopulmonary exercise (CPX) test on a cycle ergometer [6]. Interestingly, these altered responses were not associated with HbA_1c_ but this might have been due to the tight glycemic control in the T1DM cohort of this study (mean HbA_1c_ 6.9% [6.2;7.7%]). Thus, the jury is still out whether or not glucose control assessed by HbA_1c_ is associated with maximum oxygen uptake. 

In clinical practice, this question is highly relevant for physically active patients and in particular for competitive athletes with T1DM. Therefore, this systematic review investigates the association between HbA_1c_ levels and VO_2max_ assessed during CPX testing in individuals with T1DM. 

## 2. Materials and Methods

This systematic review and comprehensive analysis were conducted according to the Preferred Reporting Items for Systematic Reviews (PRISMA) guidelines. The study was registered at the International Prospective Register of Systematic Reviews (Prospero) prior to initiation of the literature search (CRD42020141164). 

### 2.1. Data Sources and Study Selection

The following electronic libraries were searched to identify relevant publications: EMBASE, PubMed, Cochrane Central Register of Controlled Trials, and ISI Web of Knowledge. Studies from January 1950 until July 2020 were included in the analysis. Search items were included as shown in Appendix A. Randomized and observational studies with a minimum number of three participants were included if HbA_1c_ measurements and CPX tests to determine VO_2max_ had been performed. Only published studies were considered. Moreover, no in silico or animal studies were included. Moreover, systematic reviews and meta-analysis were excluded. Additionally, duplicates of articles were discarded. Study titles and abstracts were reviewed including relevant studies fulfilling the inclusion criteria. Then, the full text of these studies was digitally saved and read by two independent authors (MLE, F.J.R.D.). Another independent author (O.M) monitored the identified studies and solved potential disagreements. The SD values were converted to standard error (SE = SD/√n) [7]. Interquartile range (IQR) or confidence intervals were converted using the formula provided in the Appendix A. If studies included more than one appropriate data set, these data were extracted and analyzed separately. 

### 2.2. Criteria for Inclusion in the Review

The following criteria had to be met for the study abstract to be considered eligible for manuscript data extraction: (a) reporting of an association between HbA_1c_ at baseline and VO_2max_, (b) at least three participating children aged 6–12, adolescents aged 13 to 17 or adults older than 18 years, (c) participants with type 1 diabetes, (d) observational, cross-sectional or randomized controlled study design, (e) conducted a CPX test. Studies without clear specification of diabetes type were not included.

### 2.3. Data Extraction and Quality Assessment

Following information, if available, was extracted from all studies: authors, year of publication, country of study origin, trial design, sample size, age and sex of participants and CPX-test procedure. If data were missing, authors were asked to provide these data. If the main outcome (e.g., BG delta) was not reported, could not be retrieved after contacting the authors, or computed, the study was excluded. 

Studies were independently assessed by two investigators (M.L.E., F.J.R.D.) for methodological quality using the risk of bias assessment tool from the Cochrane Collaboration [8] in its revised version [9]. The following sources of bias were detected: overall bias, selection of the reported result, bias of the measurement of the outcome, missing outcome data, deviations from the intended interventions and randomization process (Figure 1). We did not exclude any studies based on the risk of bias assessment since the included trials were judged to possess a low risk of bias following the assessment (Figure 1).

### 2.4. Data Synthesis and Analysis

A narrative descriptive analysis was performed to summarize the characteristics of studies, such as population, age and type of CPX testing. VO_2max_ was defined as the maximum oxygen consumption given in each study measured via a CPX test on a spirometric device independent of manufacturing company. HbA_1c_ values were recorded as given in the anthropometry section of the included cohorts within each manuscript. Studies were excluded if VO_2max_ values were not measured according to the guidelines of the American College of Sports Medicine [10,11].

#### Meta-Analysis

The meta-analysis was performed using the random effects model and Hedges’ g method as a number of studies only had a small sample size. The effect size (delta BG) was summarized and presented as the pooled mean with corresponding 95% confidence interval (CI). The negative pooled mean indicated a higher decrease in BG following physical exercise. The heterogeneity in the effect size was assessed by estimating I2 statistics and Cochran’s Q test for homogeneity. The difference in effect size with respect to cycling versus running studies and other study-level categorical covariates was assessed by performing sub-group analysis of the effect size for each covariate and group differences in the effect size were assessed via Cochran’s Q test for homogeneity. The difference in effect size with respect to the study-level continuous covariates was assessed by meta-regression analysis. Furthermore, simple and multiple meta-regression were performed to assess the crude and adjusted association of each study-level covariate with the effect size within the strata of cycling and treadmill studies. The results of the meta-regression were reported as a coefficient with corresponding 95% CI and *p* values. Publication bias was assessed in terms of meta-bias using Egger’s test and visualized via a funnel plot. The results of the meta-analysis are presented in the Appendix A.

## 3. Results

A total of 122 studies were extracted from the initially screened 4416 titles. The steps of the article selection process are described as a flow diagram in Figure 2. Studies published between 1989 and 2020 included data from 3278 participants with T1DM. Due to the crossover designs of the studies the results were split for analysis and led to a total number of 187 study results included. We included 57 case–control studies, 41 randomized controlled trials, 2 secondary outcome analyses from randomized controlled trials, 7 non-randomized comparative studies and 15 cohort studies. Groups were separated by adolescents, adolescents/adults and adults, since some study cohorts did not include both age-groups without a clear separation within the results of the study. Detailed information is shown in Figure 2.

All study-specific outcomes included in this systematic review and meta-analysis are shown in Table 1.

### 3.1. Primary Endpoint

The overall pooled mean values were 8.1% [95% CI: 7.9–8.3%] for HbA_1c_ and 38.5 (37.3–39.6) mL/kg/min for VO_2max_. Studies including participants with a mean HbA_1c_ ≥7.5% had a significantly lower VO_2max_ with 36.9 [35.9–38.1] mL/kg/min in comparison to those having an HbA_1c_ < 7.5 with 41.3 (39.2–43.4) mL/kg/min (*p* < 0.001). For both, VO_2max_ (I2 = 98.09%) and HbA_1c_ (I2 = 96.77%) high heterogeneity for study results was observed.

There was a weak, but significant correlation between VO_2max_ and HbA_1c_ of r = −0.19 (*p* < 0.001). The meta-regression revealed a slope between VO_2max_ and HbA_1c_ of −1.46 [−2.35–−0.58], *p* < 0.001), i.e., for each increase in HbA_1c_ of 1% the relative VO_2max_ was lower by 1.46 mL/kg/min. This association is shown in Figure 3.

### 3.2. Subgroup Analysis

Subgroup analyses were conducted for sex, age group and type of exercise. The highest pooled mean VO_2max_ was observed in studies including only male participants (44.3 [41.9–46.6] mL/kg/min vs. 32.2 (28.5–35.8) mL/kg/min in women and 36.6 (35.4–37.7) mL/kg/min in studies including both sexes, *p* < 0.001). Overall participants had a mean pooled HbA_1c_ of 8.1% (95% CI: 7.9–8.3%) with male participants at 7.8% (7.5–8.1) and female participants at 8.8% (8.2–9.5) (*p* = 0.01).

Only minor differences were observed for VO_2max_ across the age groups: adolescents had a VO_2max_ of 37.6 (35.4–39.7) mL/kg/min, adolescents/adults 38.9 (35.9–41.9) mL/kg/min, adults 38.6 (37.0–40.1) mL/kg/min and non-specified groups 38.4 (33.8–42.9) (*p* = 0.857). Only two studies reported on VO_2max_ and HbA_1c_ in children, those results are included in Appendix A under the subgroup ‘other’. In contrast, clear differences were seen for VO_2max_ for different types of exercise; cycling led to a mean pooled VO_2max_ of 36.9 (35.5–38.2) mL/kg/min vs. 43.5 (41.4–45.5) mL/kg/min with treadmill and 40.4 (37.5–43.2) mL/kg/min for other types of exercise to (*p* < 0.001).

### 3.3. Multivariate Meta-Regression

Multivariate meta-regressions for VO_2max_ in association with HbA_1c_, sex, age group and type of exercise were conducted. The results are shown in Table 2. Univariate Meta-regression can be found in the Appendix A.

## 4. Discussion

Our systematic review and meta-analysis showed that HbA_1c_ and VO_2max_ are inversely, albeit weakly associated. Furthermore, if groups were separated for HbA_1c_, significant differences in VO_2max_ between low (<7.5%) and high (≥7.5%) values were found. Additional subgroup analyses showed that VO_2max_ was highest in male (in comparison to female) participants whereas age only had a small, insignificant effect on VO_2max_ and on the impact of HbA_1c_ on VO_2max_.

Physical exercise has previously shown to not necessarily decrease HbA_1c_ for a variety of reasons [134,135]. Individuals with T1DM are advised to reduce insulin doses in preparation for specific exercise sessions, which may elevate BG and eventually (at least with regular physical exercise) HbA_1c_ [136]. In addition, patients are advised to supplement CHO with falling glucose values during exercise [136]. Even though the amounts of administered CHO are small, they may weaken the positive exercise induced effects on body mass and glycemic control.

Moreover, previous exercise studies have shown that slightly elevated BG levels or CHO supplementation during physical exercise may increase the performance of individuals without T1DM [136]. In combination with elevated pre-exercise BG values (due to insulin reductions) and higher post-exercise BG levels (in response to supplemented CHO during exercise), HbA_1c_ might be negatively impacted in those people with T1DM who are physically active, and fear of hypoglycemia might be another contributor.

However, a large multi-center study demonstrated that physical activity has a positive impact on glycemic control but also on diabetes-related comorbidities [137]. This is supported by the study from King et al. that found a positive association between glycemic control and the amount of physical activity in children with T1DM [138]. These studies are reflective of the results from our systematic review and meta-analysis that show an inverse relationship between HbA_1c_ and VO_2max_, highlighting that a 1% increase in HbA_1c_ decreases VO_2max_ by 1.46 mL/kg/min. This may appear negligible at first, but Laukkanen et al. have shown that a decrease of 1 mL/kg/min of VO_2max_ is associated with a 9% increase in all-cause mortality in a population-based study of 579 men without diabetes [139]. VO_2max_ has previously been suggested to predict longevity. While still a matter of research, an elevated VO_2max_ allows individuals to be more active which may increase well-being and longevity [140]. It should be noted that HbA_1c_ worsens between the ages of 8–18, while from the age of 16 it steadily improves due to a higher awareness of diabetes management. In addition, puberty and hormonal changes at that time contribute to a more complicated glycemic management that may ease over time [141]. Adolescents in our study were already in the ‘steadily improving’ age range hence the impact of puberty and hormonal changes are not as pronounced or influential on our study results. Furthermore, adolescents with T1DM do not necessarily have a smaller VO_2max_ compared to adults since studies in healthy individuals [142] and individuals with T1DM [143] align with our overall findings regarding VO_2max_. 

Even though the association was weak in our results, HbA_1c_ has shown to reduce skeletal muscle mitochondrial ATP production, which consequently reduces performance, deteriorating VO_2max_ [144]. In individuals with an increased HbA_1c_, capillary density around skeletal musculature has also been shown to be decreased [145]. This may in addition lead to compromised oxygen supply systems influencing an individual’s functional capacity.

These studies and our systematic review and meta-analysis are not without limitations. Although still considered the gold standard of glycemic control, HbA_1c_ may not be the ideal parameter to detect the effects of physical exercise on glucose control in individuals with T1DM as it is an average of low and high BG values over a rather long period of time. Low BG values will reduce HbA_1c_ although they are not exactly an indicator of good glycemic control [146]. On the other hand, exercise-induced hyperglycemic BG values which are often accepted during physical activity will increase HbA_1c_ which might cast doubt on the benefit of exercise in people with T1DM. Another contributor to the effects seen in our analysis could be diabetes duration. This may have an impact on HbA_1c_ but also on VO_2max_ since it may decrease over time with insufficient training. However, it might not necessarily have an impact since diabetes duration could be influential once individuals with T1DM have gained more knowledge about their own diabetes which could thus improve their overall glycemic control and engagement in physical activities increasing VO_2max_. Unfortunately, this could not be included in our systematic review and meta-analysis, since the majority of the studies did not show the diabetes duration in detail, omitting this important detail which should be considered in future studies.

Future studies should focus on more differentiated parameters than HbA_1c_ such as time in range which represents the standard parameter to assess quality of glycemic control in users of continuous glucose monitoring systems. Hypoglycemic and hyperglycemic glucose values do not compensate for each other in time in range. Therefore, time in range might be a more appropriate parameter to evaluate quality of glycemic control in particular for interventions potentially triggering hypoglycemia, including physical activity. It must be mentioned, that several confounding parameters may have an impact on HbA_1c_ without influencing VO_2max_. This could be an adaption in dietary behavior, e.g., meal timing or overall carbohydrate intake, or this could be a change in therapy from pen to pump therapy or an upgrade to a novel insulin analogue that may lower HbA_1c_ immediately. Future studies on this important research topic will hopefully investigate additional parameters potentially affecting HbA_1c_ and its association with VO_2max_, such as body mass, comorbidities and socioeconomic status—parameters that could not be included in this meta-analysis since as they were either not measured or reported insufficiently, leaving important information about the association of HbA_1c_ and VO_2max_ unexplored.

In our systematic review and meta-analysis, we chose the gold standard parameters for glycemic control and maximum oxygen uptake, HbA_1c_ and VO_2max_, since they are the most popular and most often applied parameters in diabetes research, but VO_2max_ has its limitations, too. It is unable to fully represent the entire spectrum of physical performance since cardiological (heart rate max), metabolic (lactate max) and physiological parameters such as power and speed are not reflected. Since we solely included studies with individuals that conducted exercise tests until volitional exhaustion for the determination of VO_2max_, results from our study cannot be transferred to all types of physical exercise.

Nevertheless, future studies are needed to investigate additional parameters that may impact long-term glycemic control and its association with VO_2max_ in individuals with T1DM.

## 5. Conclusions

This meta-analysis demonstrates an inverse association between physical performance and HbA_1c_ showing an increase in VO_2max_ with decreasing HbA_1c_. Further studies relating to time in range will be needed to confirm a positive impact of glucose control on physical performance.

## Figures and Tables

**Figure 1 metabolites-12-01017-f001:**
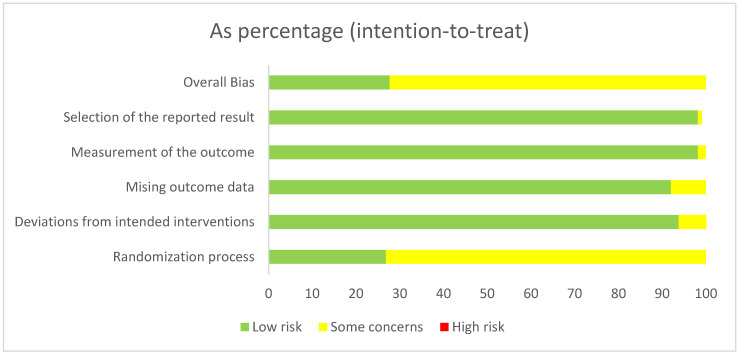
Risk of bias summary. Risk of bias was assessed according to the methods recommended by the Cochrane Collaboration [8].

**Figure 2 metabolites-12-01017-f002:**
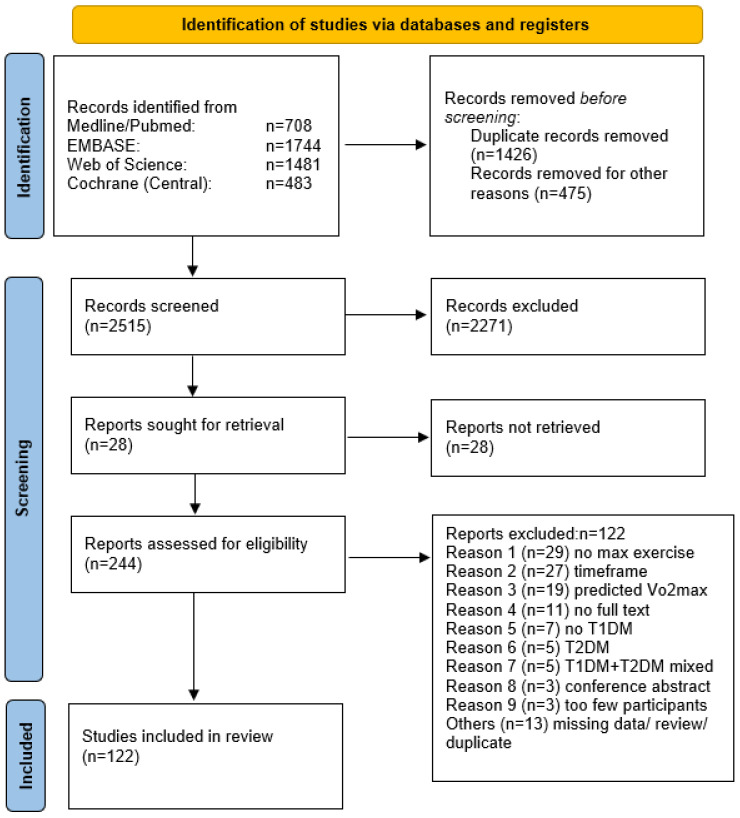
PRISMA statement [12].

**Figure 3 metabolites-12-01017-f003:**
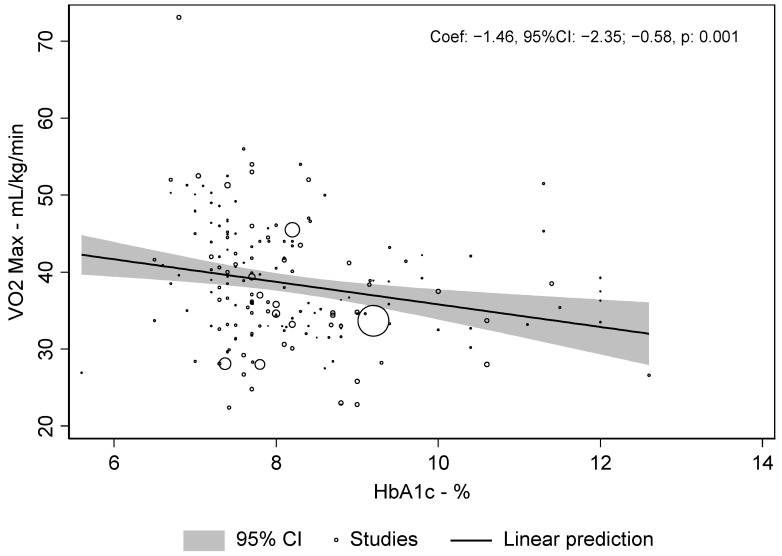
Meta-regression dot plot for HbA_1c_ and VO_2max_.

**Table 1 metabolites-12-01017-t001:** Studies included in systematic review and meta-analysis with HbA_1c_ and VO_2max_.

Title	Sex	Age-Set	Type of Exercise	HbA_1c_ ± SD (%)	VO_2max/peak_ ± SD (mL/kg/min)	Sample Size
Abraham, MB 2017 [13]	both (m/w = 4/4)	adolescents	other	7.8 ± 1	39.7 ± 6.10	8
Adolfsson, P 2012 (a) [14]	male	adolescents	cycle ergometry	7.6 ± 0.88	56.00 ± 5.50	12
Adolfsson, P 2012 (b)	female	adolescents	cycle ergometry	8.1 ± 0.58	44.00 ± 6.68	12
Adolfsson, P 2012 (c)	both	adolescents	cycle ergometry	7.9 ± 3.07	49.80 ± 9.90	12
Al Khalifah, RA (a) 2016 [15]	both (m/w = 15/8)	adults	cycle ergometry/treadmill	7.8 ± 0.9	44.00 ± 8.80	23
Al Khalifah, RA (b) 2016	both (m/w = 11/10)	adults	cycle ergometry/treadmill	7.6 ± 0.9	26.70 ± 5.00	21
Atalay, M 1997 [16]	male	adults	cycle ergometry	7.3 ± 1.7	46.00 ± 6.90	9
Austin, A 1993 [17]	both (m/w = 28/31)	adolescents	cycle ergometry	10.6 ± 2.1	33.70 ± 7.00	59
Bak, JF 1989 [18]	undefined	adults	cycle ergometry	7.9 ± 1.4	45.70 ± 7.40	7
Baldi, JC 2010 [19]	both (m/w = 80%/20%)	adults	cycle ergometry	7.3 ± 0.8	42.00 ± 8.00	12
Bally, L (1) 2016 [20]	male	adults	cycle ergometry	7 ± 0.6	47.90 ± 10.20	12
Bally, L (2) 2016 [21]	male	adults	cycle ergometry	7 ± 0.6	48.00 ± 11.20	10
Baraldi, E 1992 [22]	both (m/w = 17/16)	adolescents	treadmill	8.9 ± 1.8	41.20 ± 5.90	33
Benbassat, CA 2001 [23]	both (m/w = 9/7)	adults	cycle ergometry	8.6 ± 1.8	27.50 ± 12.00	15
Bjornstad, P 2018 (a) [24]	both (m/w = 48%/52%)	adolescents	cycle ergometry	9 ± 1.6	25.80 ± 4.60	27
Bjornstad, P 2018 (b)	both (m/w = 48%/52%)	adolescents	cycle ergometry	8.8 ± 1.4	33.00 ± 7.80	48
Bjornstad, P 2018 (c)	both (m/w = 48%/52%)	adolescents	cycle ergometry	8.2 ± 1.4	33.20 ± 4.40	52
Bjornstad, P 2018 (d)	both (m/w = 48%/52%)	adolescents	cycle ergometry	9 ± 1.6	22.80 ± 4.90	27
Bjornstad, P 2018 (e)	both (m/w = 48%/52%)	adolescents	cycle ergometry	8.8 ± 1.4	23.00 ± 5.80	48
Bjornstad, P 2018 (f)	both (m/w = 48%/52%)	adolescents	cycle ergometry	8.2 ±1.4	30.10 ± 8.00	52
Bjornstad, P 2015 [25]	both	adolescents	cycle ergometry	8.5 ± 1.4	31.50 ± 6.30	69
Boff, W 2019 (a) bl. [26]	both (m/w = 3/6)	adults	cycle ergometry	8.2 ± 1.3	34.00 ± 6.30	9
Boff, W 2019 (b) pi.	both (m/w = 3/6)	adults	cycle ergometry	8.2 ± 1.3	40.10 ± 4.30	9
Boff, W 2019 (c) bl.	both (m/w = 5/4)	adults	cycle ergometry	8.4 ± 0.9	33.00 ± 8.20	9
Boff, W 2019 (d) pi.	both (m/w = 5/4)	adults	cycle ergometry	8.4 ± 0.9	36.00 ± 8.80	9
Boff, W 2019 (e) bl.	both (m/w = 4/5)	adults	cycle ergometry	8.8 ± 2.3	33.20 ± 10.00	9
Boff, W 2019 (f) pi.	both (m/w = 4/5)	adults	cycle ergometry	8.8 ± 2.3	32.70 ± 10.00	9
Bracken, RM 2012 [27]	both (m/w = 2/5)	adults	treadmill	9.16 ± 2.74	38.90 ± 4.40	7
Bracken, RM 2011 [28]	both (m/w = 6/1)	adults	treadmill	8.3 ± 0.1	43.50 ± 0.90	7
Brazeau, AS (1a) 2012 [29]	male	adults	cycle ergometry	7.71 ± 1.25	35.90 ± 10.50	22
Brazeau, AS (1b) 2012	male	adults	cycle ergometry	7.42 ± 1.25	29.90 ± 7.70	18
Brazeau, AS (1c) 2012	female	adults	cycle ergometry	7.71 ± 1.25	28.30 ± 6.20	15
Brazeau, AS (1d) 2012	female	adults	cycle ergometry	7.42 ± 1.25	22.40 ± 5.20	20
Brazeau, AS (2a) 2012 [30]	male	adults	cycle ergometry	7.5 ± 0.9	33.10 ± 9.80	40
Brazeau, AS (2b) 2012	female	adults	cycle ergometry	7.7 ± 1.6	24.80 ± 6.30	37
Brazeau, AS (2c) 2012	both	adults	cycle ergometry	7.6 ± 1.3	29.20 ± 9.20	77
Brugnara, L 2012 [31]	male	adults	cycle ergometry	6.9 ± 1	35.00 ± 6.50	10
Bussau, VA 2006 [32]	male	adults	cycle	7.4 ± 0.8	44.50 ± 4.20	7
Bussau, VA 2007 [33]	male	adults	cycle ergometry	7.4 ± 0.7	45.20 ± 5.00	7
Campaigne, BN 1987 [34]	male	adults	cycle ergometry	7.4 ± 0.3	36.60 ± 1.60	9
Campbell, MD 2013 [35]	male	adults	treadmill	7.7 ± 0.3	53.00 ± 1.00	11
Campbell, MD (1) 2014 [36]	male	adults	treadmill	7.7 ± 0.4	54.00 ± 1.00	8
Campbell, MD (2) 2014 [37]	male	adults	treadmill	6.7 ± 0.7	52.00 ± 4.00	10
Campbell, MD (1) 2015 [38]	both (m/w = 7/2)	adults	treadmill	8.1 ± 0.2	41.80 ± 1.60	9
Campbell, MD (2) 2015 [39]	male	adults	treadmill	6.9 ± 0.2	51.30 ± 2.10	10
Chokkalingam, K 2007 [40]	male	adults	cycle ergometry	7.9 ± 0.2	44.50 ± 1.20	8
de Jesus, IC 2019 [41]	both (m/w = 5/4)	adolescents	cycle ergometry	9.39 ± 1.25	38.79 ± 10.02	9
de Lima, VA 2017 [42]	both (m/w = 25/20)	adolescents	cycle ergometry	9.15 ± 1.61	38.38 ± 7.54	45
D’hooge, R (a) 2011 bl. [43]	both	adolescents	cycle ergometry	8.13 ± 1.02	32.87 ± 7.83	8
D’hooge, R (b) 2011 pi.	both	adolescents	cycle ergometry	8.08 ± 0.97	32.99 ± 9.58	8
D’hooge, R (c) 2011 bl.	both	adolescents	cycle ergometry	8.55 ± 0.82	35.19 ± 6.71	8
D’hooge, R (d) 2011 pi.	both	adolescents	cycle ergometry	8.48 ± 0.9	34.69 ± 9.24	8
Dovc, K (a) 2017 [44]	male	adolescents	cycle ergometry	7.5 ± 0.5	49.2 ± 8.1	11
Dovc, K (b) 2017	female	adolescents	cycle ergometry	7.9 ± 0.7	36.1 ± 4	9
Dovc, K (c) 2017	both	adolescents	cycle ergometry	7.7 ± 0.6	43.3 ± 9.3	20
Ebeling, P (a) 1995 [45]	undefined	adults	cycle ergometry	8.4 ± 0.4	52.0 ± 1.0	11
Ebeling, P (b) 1995	undefined	adults	cycle ergometry	7.2 ± 0.2	42.0 ± 1.0	12
Farinha, JB (a) 2018 bl. [46]	both (m/w = 5/4)	adults	cycle ergometry	7.5 ± 1.5	31.3 ± 6.0	9
Farinha, JB (b) 2018 pi.	both (m/w = 5/4)	adults	cycle ergometry	7.2 ± 1.1	37.4 ± 8.7	9
Farinha, JB (c) 2018 bl.	both (m/w = 5/4)	adults	cycle ergometry	8.1 ± 1.3	32.4 ± 6.3	9
Farinha, JB (d) 2018 pi.	both (m/w = 5/4)	adults	cycle ergometry	8 ± 0.8	34.3 ± 5.2	9
Farinha, JB (e) 2018 bl.	both (m/w = 5/5)	adults	cycle ergometry	7.5 ± 1	31.4 ± 7.1	10
Farinha, JB (f) 2018 pi.	both (m/w = 5/5)	adults	cycle ergometry	7.2 ± 0.7	33.0 ± 7.8	10
Faulkner, MS (a) 2010 bl. [47]	both (m/w = 9/3)	adolescents	cycle ergometry	9.4 ± 1.8	33.3 ± 6.9	12
Faulkner, MS (b) 2010 pi.	both (m/w = 9/3)	adolescents	cycle ergometry	9.4 ± 2.1	35.8 ± 8.8	12
Faulkner, MS 2005 [48]	both (m/w = 57/48)	adolescents	cycle ergometry	8.7 ± 1.6	34.4 ± 8.8	105
Fintini, D 2012 [49]	both (m/w = 15/20)	children	treadmill	7.7 ± 0.8	36.2 ± 7.4	35
Franc, S 2015 [50]	both (m/w = 11/9)	adults	cycle ergometry	7.9 ± 0.9	33.0 ± 10.0	20
Francis, SL (a) 2015 [51]	male	adolescents	treadmill	8.6 ± 0.9	50.0 ± 6.5	10
Francis, SL (b) 2015	female	adolescents	treadmill	8.2 ± 0.9	44.0 ± 6.3	10
Francis, SL (c) 2015	both	adolescents	treadmill	8.4 ± 0.7	47.0 ± 6.9	20
Fuchsjager-Mayrl, G 2002 (a) bl. [52]	both (m/w = 7/11)	adults	cycle ergometry	7.3 ± 0.2	28.1 ± 1.2	18
Fuchsjager-Mayrl, G 2002 (b) pi.(1)	both (m/w = 7/11)	adults	cycle ergometry	7.7 ± 0.3	31.8 ± 2	18
Fuchsjager-Mayrl, G 2002 (c) pi.(2)	both	adults	cycle ergometry	7.5 ± 0.3	35.7 ± 2.8	15
Fuchsjager-Mayrl, G 2002 (d) pi.(3)	both	adults	cycle ergometry	7 ± 0.2	28.4 ± 1.8	13
Fuchsjager-Mayrl, G 2002 (e) bl.	both	adults	cycle ergometry	7.4 ± 0.4	29.6 ± 2.3	8
Fuchsjager-Mayrl, G 2002 (f) pi.	both	adults	cycle ergometry	7.4 ± 0.2	29.7 ± 2.4	8
Giani, E 2018 [53]	both (m/w = 53%/47%)	adolescents	cycle ergometry	7.4 ± 1.0	33.2 ± 6.2	17
Goulding, R 2020 [54]	male	adults	cycle ergometry	7.3 ± 0.9	36.4 ± 4.7	17
Gray, BJ 2016 [55]	both (m/w = 2/5)	adults	treadmill	9.2 ± 0.6	38.9 ± 4.4	7
Guelfi, KJ 2005 [56]	both (m/w = 4/3)	adults	other	7.4 ± 1.5	39.4 ± 7.4	7
Guelfi, KJ 2007 [57]	both (m/w = 5/4)	adults	cycle ergometry	7.7 ± 0.8	41.8 ± 4.6	9
Gusso, S 2008 [58]	female	adolescents	cycle ergometry	8.8 ± 0.3	31.6 ± 2	12
Gusso, S 2012 [59]	both (m/w = 27/26)	adolescents	cycle ergometry	8.7 ± 0.2	33.1 ± 1.0	53
Haagglund, H 2012 [60]	male	adults	cycle ergometry	7.7 ± 0.9	36 ± 4	10
Heise, T 2016 [61]	both (m/w = 35/5)	adults	other	7.7 ± 0.8	39.4 ± 3.7	40
Heyman, E 2020 [62]	both (m/w = 12/4)	adults	cycle ergometry	8.3 ± 1.5	34.9 ± 7.2	16
Heyman, E 2007 [63]	female	adolescents	cycle ergometry	8.1 ± 1.3	30.6 ± 4.0	19
Hilberg, T 2004 [64]	male	adults	cycle ergometry	7.2 ± 0.2	49.0 ± 2.2	16
Jenni, S 2008 [65]	male	adults	cycle ergometry	6.7 ± 0.2	50.3 ± 4.5	7
Jensen, T 1988 (a) [66]	both (m/w = 6/4)	adults	cycle ergometry	7.5 ± 0.95	40.7 ± 9.5	10
Jensen, T 1988 (b)	both (m/w = 6/4)	adults	cycle ergometry	8.7 ± 1.21	28.4 ± 8.8	10
Jensen, T 1988 (c)	both (m/w = 6/4)	adults	cycle ergometry	9.3 ± 1.0	28.2 ± 4.2	10
Komatsu, WR 2010 (a) [67]	undefined	adults	treadmill	7.5 ± 6.2	42.4 ± 5.5	15
Komatsu, WR 2010 (b)	undefined	adults	treadmill	9 ± 1.3	34.8 ± 3.3	12
Komatsu, WR 2005 [68]	both (m/w = 38/34)	adolescents	treadmill	8.1 ± 2.2	41.6 ± 7.7	72
Koponen, AS 2013 [69]	male	adults	cycle ergometry	7.65 ± 0.8	35.4 ± 4.8	12
Kornhauser, C 2012 [70]	both (m/w = 5/5)	adolescents	treadmill	10 ± 1.0	37.5 ± 2.7	10
Laaksonen, DE 2000 (a) bl. [71]	male	adults	cycle ergometry	8.2 ± 1.1	43.4 ± 8.0	20
Laaksonen, DE 2000 (b) pi.	male	adults	cycle ergometry	8 ± 1.0	46.1 ± 6.6	20
Laaksonen, DE 1996 [72]	male	adults	cycle ergometry	7.3 ± 1.7	46 ± 6.9	9
Landt, KW 1985 (a) bl. [73]	both (m/w = 3/6)	adolescents	cycle ergometry	12 ± 1.0	36.3 ± 3	9
Landt, KW 1985 (b) pi.	both (m/w = 3/6)	adolescents	cycle ergometry	12 ± 1.0	39.3 ± 3	9
Landt, KW 1985 (c) bl.	both (m/w = 4/2)	adolescents	cycle ergometry	12 ± 1.0	39.2 ± 3.4	6
Landt, KW 1985 (d) pi.	both (m/w = 4/2)	adolescents	cycle ergometry	12 ± 1.0	37.5 ± 3.3	6
Lee, MJ 2016 [74]	both (m/w = 45.8%/54.2%)	(adolescents)/adults	cycle ergometry	7.9 ± 1.3	34.9 ± 5.8	24
Lehmann, R 1997 (a) bl. [75]	both (m/w = 13/7)	adults	cycle ergometry	7.6 ± 1.0	41.2 ± 13.1	20
Lehmann, R 1997 (b) pi.	both (m/w = 13/7)	adults	cycle ergometry	7.5 ± 0.9	45.0 ± 13.2	20
Matthys, D 1996 [76]	both (m/w = 12/18)	adolescents	cycle ergometry	10 ± 0.3	32.5 ± 2.1	30
McCarthy, O 2020 [77]	male	adults	cycle ergometry	6.8 ± 0.6	73.1 ± 3.8	16
McKewen, MW 1999 [78]	male	adults	cycle ergometry	7.2 ± 1.2	50.3 ± 7.4	7
Michaliszyn, SF 2009 [79]	both (m/w = 60/49)	adolescents	other	8.7 ± 1.6	34.7 ± 8.9	109
Moser, O 2017 [80]	both (m/w = 51/13)	adults	cycle ergometry	7.8 ± 1.0	37.0 ± 5.0	64
Moser, O 2019 [81]	both (m/w = 5/4)	adults	cycle ergometry	7.2 ± 0.6	39.0 ± 12.0	9
Moser, O 2018 (1) [82]	male	adults	cycle ergometry	7.4 ± 0.6	52.5 ± 6.6	7
Moser, O 2018 (2) [83]	both (m/w = 51/13)	adults	cycle ergometry	7.8 ± 1.0	37.0 ± 5.0	64
Murray, FT 1988 [84]	male	adults	other	12 ± 0.6	33.5 ± 2.6	8
Nadeau, KJ 2010 [85]	both (m/w = 6/6)	adolescents	cycle ergometry	8.65 ± 1.6	31.5 ± 7.6	12
Nguyen, T 2015 (a) [86]	both (m/w = 5/3)	adolescents	cycle ergometry	7.4 ± 0.5	38.5 ± 5.8	8
Nguyen, T 2015 (b)	both (m/w = 5/3)	adolescents	cycle ergometry	11.1 ± 1.0	33.2 ± 5.6	8
Niranjan, V 1997 (a) [87]	both (m/w = 7/2)	adults	cycle ergometry	5.6 ± 0.2	26.9 ± 2.6	9
Niranjan, V 1997 (b)	both (m/w = 4/5)	adults	cycle ergometry	8.8 ± 0.5	22.8 ± 3.5	9
Peltonen, JE 2012 [88]	male	adults	cycle ergometry	7.7 ± 0.7	34.7 ± 4.4	10
Peltoniemi, P 2001 [89]	male	adults	cycle ergometry	7 ± 0.3	45.0 ± 2.0	12
Poortmans, JR 1986 (a) [90]	male	adolescents	cycle ergometry	7.3 ± 0.3	40.6 ± 1.3	9
Poortmans, JR 1986 (b)	male	adolescents	cycle ergometry	11.4 ± 0.9	38.5 ± 1.0	8
Raguso, CA 1995 [91]	male	adults	cycle ergometry	8 ± 0.7	40.5 ± 2.0	7
Reddy, R 2019 [92]	both (m/w = 4/6)	adults	treadmill	7.4 ± 1.0	46.8 ± 11.6	10
Rigla, M 2000 (a) bl. [93]	both (m/w = 7/7)	adults	treadmill	6.5 ± 0.8	33.7 ± 7.0	14
Rigla, M 2000 (b) pi.	both (m/w = 7/7)	adults	treadmill	6.7 ± 1.0	38.5 ± 7.7	14
Rigla, M 2001 (a) bl. [94]	both (m/w = 7/7)	adults	other	6.5 ± 0.8	33.7 ± 7.0	14
Rigla, M 2001 (b) pi.	both (m/w = 7/7)	adults	other	6.7 ± 1.0	38.5 ± 7.7	14
Rissanen, APE 2015 [95]	male	adults	cycle ergometry	7.4 ± 0.9	40.0 ± 3.0	7
Rissanen, APE 2018 (a) bl. [96]	male	adults	cycle ergometry	7.3 ± 0.9	38.0 ± 4.0	8
Rissanen, APE 2018 (b) pi.	male	adults	cycle ergometry	7.5 ± 1.1	41.0 ± 3.0	8
Roberts, TJ 2018 (1) [97]	both (m/w = 29/11)	adults	cycle ergometry	7.7 ± 1.3	32.0 ± 10.0	40
Roberts, TJ 2018 (2) [98]	both (m/w = 13/7)	adults	cycle ergometry	8.1 ± 3.9	38.0 ± 9.0	20
Roberts, TJ 2020 [99]	both (m/w = 24/10)	adults	cycle ergometry	7.8 ± 1.3	33.0 ± 10.0	34
Robitaille, M 2007 [100]	both (m/w = 5/3)	adults	cycle ergometer	7.4 ± 0.4	42.9 ± 10.3	8
Roche, DM 2008 (a) [101]	male	adolescents	treadmill	9.4 ± 1.0	43.2 ± 7.3	15
Roche, DM 2008 (b)	female	adolescents	treadmill	9.8 ± 1.7	39.2 ± 9.0	14
Roche, DM 2008 (c)	both	adolescents	treadmill	9.6 ± 1.4	41.4 ± 8.2	29
Rowland, TW 1992 [102]	male	adolescents	cycle ergometry	11.3 ± 3.0	51.5 ± 5.8	11
Roy-Fleming, A 2019 [103]	both (m/w = 11/11)	adults	cycle ergometry	7.3 ± 1.0	32.6 ± 7.1	22
Sandoval, DA 2004 [104]	both (m/w = 14/13)	adults	cycle ergometry	7.8 ± 0.2	28.0 ± 2.0	27
Schneider, SH 1992 (a) [105]	both (m/w = 12/4)	adults	cycle ergometry	12.6 ± 0.8	26.6 ± 1.5	16
Schneider, SH 1992 (b)	both (m/w = 25/14)	adults	cycle ergometry	11.5 ± 0.6	35.4 ± 1.9	39
Seeger, JPH 2011 [106]	both (m/w = 4/5)	children	treadmill	7.9 ± 0.6	44.0 ± 5.9	9
Shetty, VB 2018 [107]	both (m/w = 4/4)	adults	cycle ergometry	8.0 ± 0.7	34.5 ± 10.9	8
Singhvi, A 2014 [108]	both (m/w = 47%/53%)	adolescents	treadmill	8.42 ± 0.9	46.6 ± 6.8	20
Stettler, C 2005 [109]	male	adults	cycle ergometry	7.4 ± 0.7	44.9 ± 8.0	8
Stewart, CJ 2017 [110]	male	adults	other	7.4 ± 0.4	51.3 ± 2.2	10
Tagougui, S (1a) 2015 [111]	male	adults	cycle ergometry	6.6 ± 0.7	40.9 ± 9.3	11
Tagougui, S (1b) 2015	both (m/w = 7/5)	adults	cycle ergometry	9.1 ± 0.7	34.6 ± 7.2	12
Tagougui, S (2a) 2015 [112]	both (m/w = 7/1)	adults	other	6.8 ± 0.7	39.6 ± 8.5	8
Tagougui, S (2b) 2015	both (m/w = 6/4)	adults	other	9.0 ± 0.7	34.6 ± 7.1	10
Tagougui, S 2020 [113]	both (m/w = 20/10)	adolescents/adults	treadmill	7.6 ± 1.0	38.9 ± 10.7	30
Tonoli, C 2015 [114]	both (m/w = 8/2)	adults	cycle ergometry	7.0 ± 0.2	52.5 ± 2.7	10
Trigona, B 2010 [115]	both (m/w = 17/15)	adolescents	treadmill	8.2 ± 0.2	45.5 ± 1.44	32
Tuominen, JA 1997 [116]	both (m/w = 6/1)	adults	cycle ergometry	7.7 ± 0.3	46.0 ± 1.0	7
Turinese, I 2017 [117]	both (m/w = 13/4)	adults	cycle ergometry	7.4 ± 0.1	28.1 ± 1.3	17
Tuttle, KR 1988 [118]	both (m/w = 8/5)	adults	cycle ergometry	8.8 ± 1.6	36.4 ± 5.9	13
Valletta, JJ 2014 [119]	both (m/w = 11/12)	adults	treadmill	7.7 ± 1.3	39.9 ± 8.4	23
Veves, A 1997 (a) [120]	both (m/w = 20/3)	adults	treadmill	8.3 ± 1.4	54.0 ± 8.1	23
Veves, A 1997 (b)	both (m/w = 4/3)	adults	treadmill	9.8 ± 1.2	42.2 ± 11.6	7
Veves, A 1997 (c)	both (m/w = 11/7)	adults	treadmill	8.9 ± 1.5	36.7 ± 9.6	5
Waclawovsky, G 2016 [121]	male	adults	cycle ergometry	7.7 ± 0.2	37.1 ± 1.4	14
Wallberg-Henriksson, H 1982 (a) bl. [122]	male	adults	cycle ergometry	10.4 ± 0.7	42.1 ± 2.1	9
Wallberg-Henriksson, H 1982 (b) pi.	male	adults	cycle ergometry	11.3 ± 0.5	45.3 ± 2.2	9
Wallberg-Henriksson, H 1986 (a) bl. [123]	female	adults	cycle ergometry	10.4 ± 0.6	30.2 ± 2.1	6
Wallberg-Henriksson, H 1986 (b) pi.	female	adults	cycle ergometry	10.4 ± 0.6	32.7 ± 2.1	6
Wallberg-Henriksson, H 1986 (c) bl.	female	adults	cycle ergometry	10.6 ± 0.6	28.0 ± 0.8	7
Wallberg-Henriksson, H 1986 (d) pi.	female	adults	cycle ergometry	10.6 ± 0.6	28.0 ± 0.8	7
Wanke, T 1992 [124]	both (m/w = 31/5)	adults	cycle ergometry	9.2 ± 2.7	33.7 ± 0.7	36
West, DJ 2011 (a) [27]	both (m/w = 7/1)	adults	treadmill	8 ± 0.2	35.8 ± 0.6	8
West, DJ 2011 (b)	both (m/w = 7/1)	adults	treadmill	8 ± 0.2	34.6 ± 0.5	8
Wilson, LC 2017 [125]	both (m/w = 12/11)	adults	cycle ergometry	8.4 ± 3.7	32.0 ± 9.0	23
Yardley, JE 2012 [126]	both (m/w = 10/2)	adolescents/adults	treadmill	7.1 ± 1.1	51.2 ± 10.8	12
Yardley, JE 2013 (1) [127]	both (m/w = 10/2)	adults	treadmill	7.1 ± 1.1	51.2 ± 10.8	12
Yardley, JE 2013 (2) [128]	both (m/w = 10/2)	adults	other	7.1 ± 1.1	51.2 ± 10.8	12
Yardley, JE 2013 (3a) [129]	both	adolescents/adults	cycle ergometry/running	7.2 ± 1.2	46.4 ± 10.1	9
Yardley, JE 2013 (3b)	both	adolescents/adults	cycle ergometry/running	7.3 ± 1.1	48.6 ± 7.8	10
Zaharieva, DP 2019 [130]	both (m/w = 4/13)	adults	treadmill	6.5 ± 0.5	41.6 ± 5.9	17
Zaharieva, DP 2016 [131]	both (m/w = 5/8)	adults	treadmill	7.4 ± 0.8	46.6 ± 12.7	13
Zaharieva, DP 2017 [132]	both (m/w = 6/6)	adults	treadmill	7 ± 0.9	50.1 ± 13.7	12
Zebrowska, A 2018 (a) [133]	undefined	adults	cycle ergometry	7.2 ± 0.41	43.9 ± 7.8	14
Zebrowska, A 2018 (b)	undefined	adults	cycle ergometry	7.2 ± 0.41	40.3 ± 7.3	14

Types of exercise identified as “other” were conducted outside on a track with a spirometric wireless device or if the type of exercise during CPX was not specified.

**Table 2 metabolites-12-01017-t002:** Multivariate meta-regression of study outcomes.

	Coefficient [95% CI]	*p* Value
**HbA_1c_ (continuous)**	−0.78 [−1.56–−0.003]	0.049
**Gender**		
Both	Reference	
Female	−2.42 [−5.93–1.09]	0.176
Male	9.21 [7.03–11.4]	<0.001
**Age**		
Other	Reference	
Adolescents	−0.51 [−5.36–4.33]	0.836
Adolescents/Adults	0.75 [−4.43–5.93]	0.777
Adults	−2.46 [−7.15–2.23]	0.304
**Exercise type**		
Other	Reference	
Cycle ergometer	−3.84 [−7.18–−0.5]	0.024
Treadmill	4.24 [0.48–8.0]	0.027

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
