# Peer review of "Association of HbA1c with VO2max in Individuals with Type 1 Diabetes: A Systematic Review and Meta-Analysis"

_metabolites, 2022, doi:10.3390/metabo12111017_

Round 1

Reviewer 1 Report

Dear authors,

Thank you for an interesting and well written review of the association between glycemic control determined as HbA1c and Vo2max in people with type 1 diabetes.

I have a few comments to the manuscript:

1.      Please discuss what differences there could be between adolescents and adults in aerobic capacity? Could there be hormonal differences of interest

2.      The first 2½ lines of the Introduction section should be deleted

3.      The first 3½ lines of the Discussion section should be deleted

4.      Please discuss what central and peripheral factors in the oxygen uptake that matters in the association between HbA1c and Vo2max, even though the association was weak. And, could the there be a pulmonary limiting factor also?

Author Response

Reviewer 1

Dear authors,

Thank you for an interesting and well written review of the association between glycemic control determined as HbA1c and Vo2max in people with type 1 diabetes.

Thank you for your comment. We really do appreciate taking your time to review our manuscript. We have responded one by one to all of your comments and hope this is in agreement with your suggestions.

I have a few comments to the manuscript:

  1. Please discuss what differences there could be between adolescents and adults in aerobic capacity? Could there be hormonal differences of interest

Thank you for your comment. We have discussed differences between adolescents and adults in aerobic capacity and potentials of hormonal differences that could have an impact in the discussion of the manuscript

Page 15, Line 66 – 74

“It should be considered HbA1c worsens between 8-18 year old’s, while from the age of 16 it steadily improves due to a higher awareness of diabetes management. In addition, puberty and hormonal changes at that time contribute to a more complicated glycemic management that may ease over time [144]. Adolescents in our study were already in the ‘steadily improving’ age range hence the impact of puberty and hormonal changes are not as pronounced and influential for our study results. Furthermore, adolescents with T1DM do not necessarily have a smaller VO2max compared to adults since studies in healthy individuals [145] and individuals with T1DM [146] align with our overall findings regarding VO2max.”

  1. The first 2½ lines of the Introduction section should be deleted

Thank you for your comment. Deleted

  1. The first 3½ lines of the Discussion section should be deleted

Thank you for your comment. Deleted

  1. Please discuss what central and peripheral factors in the oxygen uptake that matters in the association between HbA1c and Vo2max, even though the association was weak. And, could the there be a pulmonary limiting factor also?

Thank you for your comment. This has been added to the discussion.

Page 15, Line 75 – 79

“Even though the association was weak in our results, HbA1c has shown to reduce skeletal muscle mitochondrial ATP production, which consequently reduces performance deteriorating VO2max [147]. In individuals with an increased HbA1c, capillary density around skeletal musculature was also shown to be decreased [148]. This may in addition lead to compromised oxygen supply systems influencing functional capacity.“

Reviewer 2 Report

This is a well-written/performed analysis on the effects of the diabetic state and metabolic control (based Hba1c) on VO2max. Results are expected based on the effects of poor metabolic control on O2 kinetics. However, from a clinical perspective, the results are relevant.

Minor comments:

1-Lines 70-74. State relevance or delete. 

2-Line 98 (section 2.2). Add CPX.

3-Line 116 (section 2.4). Authors may consider adding whether analysis data included how VO2max measurements were obtained (indirect calorimetry vs HR). This can be included under Table 1 or in the Methods.

4-Table 1, under "exercise type/entity". State at the bottom of the table what other exercise entities were used in these prior studies and indicate if VO2 data was equivalent to the typical running/cycling data..

5-Discussion, (lines 32-36). Are these statements relevant to the paper? 

Major comments:

1-The duration of diabetes is not addressed.

2-The Discussion needs to focus on the effects of HBa1c on VO2, and beyond the acute effects of exercise on blood glucose. In fact, what are the chronic changes caused by dysregulated blood glucose homeostasis on VO2? Since VO2 is dependent on both central and peripheral factors, how does an elevated HBa1c alter lung/heart/muscle function? This does not have to addressed in detail but at least mention that poor metabolic control alters function.

Author Response

Reviewer 2

This is a well-written/performed analysis on the effects of the diabetic state and metabolic control (based Hba1c) on VO2max. Results are expected based on the effects of poor metabolic control on O2 kinetics. However, from a clinical perspective, the results are relevant.

Thank you very much for taking your time to carefully read through our manuscripts and add valuable comments. We have responded to every comment and hope this is agreement with your suggestions.

Minor comments:

1-Lines 70-74. State relevance or delete. 

Thank you for your comment. Deleted.

2-Line 98 (section 2.2). Add CPX.

Thank you for your comment. We apologize for not adding it in the first place. This has been added:

Section: 2.2 Line 96-99

“Following criteria had to be met for the study abstract being considered as eligible for manuscript data extraction: a) reporting of an association between HbA1c at baseline and VO2max, b) at least 3 participating children, adolescents or adults, c) participants with type 1 diabetes, d) observational, cross-sectional or randomized controlled study design, e) conduction of a CPX test.”

3-Line 116 (section 2.4). Authors may consider adding whether analysis data included how VO2max measurements were obtained (indirect calorimetry vs HR). This can be included under Table 1 or in the Methods.

Thank you for your comment. This has been added in the Methods section under 2.4 and below Table 1.

Line 114-124

A narrative descriptive analysis was performed to summarize the characteristics of studies such as population, age and type of CPX testing. VO2max was defined as the maximum oxygen consumption given in each study measured via a CPX-test on a spirometric device independent of manufacturing company. HbA1c values were recorded as given in the anthropometry section of the included cohorts within each manuscript. Studies were excluded if VO2max values were not measured according to the guidelines of the American College of Sports Medicine [11,12].

4-Table 1, under "exercise type/entity". State at the bottom of the table what other exercise entities were used in these prior studies and indicate if VO2 data was equivalent to the typical running/cycling data..

Thank you for your comment. This has been added. In a addition the column of exercise entity has been deleted, since it was not offering additional information to the already very large table.

Page 12

“Types of exercise identified as “other” were conducted outside on a track with spirometric wireless device or if type of exercise during CPX not specified.”

5-Discussion, (lines 32-36). Are these statements relevant to the paper? 

Thank you for your comment. Those are not relevant and have been deleted.

Major comments:

1-The duration of diabetes is not addressed.

Thank you for adding this comment. We agree that this is an important point that should be addressed. We have considered it for conclusion, which unfortunately was not possible. This was due to the fact that almost half of all studies did not include diabetes duration in their manuscript. Furthermore, it was not the primary outcome of our systematic review and meta-analysis. Nevertheless, this is an important factor that should be included in future reviews investigating this matter. We have added this as a limitation in the discussion section.

Page 15 Line 74 to 82

“Another contributor to the effects seen in our analysis could be diabetes duration. This may have an impact on HbA1c but also on VO2max since it may decrease over time with insufficient training. Apart from that it must not necessarily have an impact since diabetes duration could be influential once individuals with T1DM have gained more knowledge about their own diabetes which could improve overall glycemic control and engagement in physical activities increasing VO2max. Unfortunately, this could not be included in our systematic review and meta-analysis, since the majority of studies did not show the diabetes duration in detail omitting this important detail which should be considered in future studies.”

2-The Discussion needs to focus on the effects of HBa1c on VO2, and beyond the acute effects of exercise on blood glucose. In fact, what are the chronic changes caused by dysregulated blood glucose homeostasis on VO2? Since VO2 is dependent on both central and peripheral factors, how does an elevated HBa1c alter lung/heart/muscle function? This does not have to addressed in detail but at least mention that poor metabolic control alters function.

Thank you for your comment. We have added these circumstances also in regard to adolescents in detail.

Page 15 Line 66-79

“It should be considered HbA1c worsens between 8-18 year old’s, while from the age of 16 it steadily improves due to a higher awareness of diabetes management. In addition, puberty and hormonal changes at that time contribute to a more complicated glycemic management that may ease over time [144]. Adolescents in our study were already in the ‘steadily improving’ age range hence the impact of puberty and hormonal changes are not as pronounced and influential for our study results. Furthermore, adolescents with T1DM do not necessarily have a smaller VO2max compared to adults since studies in healthy individuals [145] and individuals with T1DM [146] align with our overall findings regarding VO2max.

Even though the association was weak in our results, HbA1c has shown to reduce skeletal muscle mitochondrial ATP production, which consequently reduces performance deteriorating VO2max [147]. In individuals with an increased HbA1c, capillary density around skeletal musculature was also shown to be decreased [148]. This may in addition lead to compromised oxygen supply systems influencing functional capacity.”

Reviewer 3 Report

Thank you for giving me the opportunity to review this manuscript. The aim of the study is to verify the relationship between HbA1c and VO2max in patients with T1D.

I report here below some comments:

First of all, the article doesn't seem suitable for the journal (topics of interest of the journal).

Parts of the orginal "MDPI Template" are still present in the manuscript

--  1. Introduction

Interventional ……………..

-- Materials and Methods

Research manuscripts reporting ……. publication. 

The introduction is unclear and does not fully describe the topic addressed.

In fact, some statements do not seem correct per se, e.g. line 45

Ultimately, even considering the characteristics of the parameters considered, from the introduction as written it is not clear whether the study is necessary.

Overall, the review question was not clearly defined in terms of population.

Young patients, and in particular, those who play sports may have particular difficulties in glycemic control before, during and after sport activity. In this sense, the study does not appear to have been conducted in a way to properly manage this condition.

In the article, it refers to the registration in PROSPERO but the page has not been updated. Furthermore, amendments have been made during the review which should be described and commented on.

Why was the minimum number of 3 participants chosen?

What about the criteria for inclusion in the review?

Criterion a) why HbA1c at baseline?

Criterion b) What age range do children and adolescents have?

How many groups were considered? In particular, what about the children group (see Tab. 1; Tab. 2)?

What were the values of the HbA1c of the different age groups?

All this means that the conclusions of the study are not properly supported.

The limitations of this review should be included in the discussion section.

Author Response

Reviewer 3

Round 2

Reviewer 2 Report

The authors have addressed my comment.